# Effect of Online Home-Based Resistance Exercise Training on Physical Fitness, Depression, Stress, and Well-Being in Middle-Aged Persons: A Pilot Study

**DOI:** 10.3390/ijerph20031769

**Published:** 2023-01-18

**Authors:** Naoki Kikuchi, Takahisa Ohta, Yuto Hashimoto, Yukina Mochizuki, Mika Saito, Ayumu Kozuma, Minoru Deguchi, Takamichi Inoguchi, Maho Shinogi, Hiroki Homma, Madoka Ogawa, Koichi Nakazato, Takanobu Okamoto

**Affiliations:** 1Faculty of Sport Science, Nippon Sport Science University, Tokyo 158-8508, Japan; 2Graduate School of Physical Education Sports Science, Nippon Sport Science University, Tokyo 158-8508, Japan; 3Research Institute for Sport Science, Nippon Sport Science University, Tokyo 158-8508, Japan; 4Integrated Research Initiative for Living Well with Dementia, Tokyo Metropolitan Institute for Geriatrics and Gerontology, Tokyo 173-0015, Japan; 5Laboratory of Health and Sports Sciences, Meiji Gakuin University, Yokohama 244-8539, Japan; 6Faculty of Sociology, Kyoto Sangyo University, Kyoto 603-8555, Japan; 7Faculty of Medical Science, Nippon Sport Science University, Yokohama 227-0033, Japan

**Keywords:** online home-based training, low-load resistance exercise, mental health, well-being, fitness

## Abstract

Purpose: This study investigated the effect of online home-based resistance exercise training on fitness, depression, stress, and well-being. A total of 67 individuals participated. Of them, 28 participants (13 men and 15 women, average age: 45.1 ± 12.2 years) performed the same exercise training online (*n* = 17), using Zoom, or in person (*n =* 11) in 2020 (Study 1). In addition, 39 participants (15 men and 24 women; average age: 47.6 ± 10.8 years) performed eight weeks of online home-based resistance exercise training in 2021 (Study 2). The participants performed low-load resistance exercises twice a week for eight weeks (16 sessions). Muscle strength, thigh muscle cross-sectional area, fitness parameters, blood pressure, mental health (Center for Epidemiologic Studies-Depression Scale—CES-D; and Kessler Psychological Distress scale—K6), and well-being (Well-Being Index—WHO-5) were measured pre-and post-resistance training. In Study 1, eight weeks of online home-based resistance training improved CES-D (*p* = 0.003), and a similar tendency was observed in resistance training (RT) with the in-person group (*p* = 0.06). There was a significant improvement in CES-D symptoms after the online home-based resistance training in Study 2 (*p* = 0.009). However, there were no significant changes in the WHO-5 and K6. Our results suggest that online low-load resistance training improves fitness parameters and curbs depressive status.

## 1. Introduction

The Coronavirus Disease 2019 (COVID-19) pandemic has imposed restrictions on the lifestyles of individuals, which has led to decreased physical activity and has had a corresponding effect on mental health and well-being. Physical activity, such as step counts, decreased worldwide after COVID-19 was declared a global pandemic [1]. A previous study reported that aerobic exercise improves mental health and subjective well-being in university students [2] and patients with major depressive disorder [3]. Additionally, Taspinar et al. [4] suggested yoga and resistance training had positive effects on mental health and well-being in sedentary adults. These studies establish the effectiveness of aerobic exercise and other exercises, such as resistance exercise, on mental health and well-being [5].

Resistance exercise is recommended by the World Health Organization (WHO) as a part of its guidelines on physical activity. The WHO recommends resistance exercise not only because it can be completed indoors (irrespective of weather) but also because it attenuates the risk of falls through enhanced muscle strength [6]. A previous meta-analysis of randomized clinical trials showed that resistance exercise could be considered therapeutic for depressive symptoms [7]. Supervised resistance exercise, especially, was more efficient in the treatment of depressive symptoms compared with unsupervised resistance exercise.

Therefore, it is necessary to clarify the impact of online participation in resistance exercise (non-contact and interactive) on mental health and physical fitness. Online body-mass-based exercise has become one of the novel exercise modes during the COVID-19 pandemic. Previous studies have shown that body-mass-based exercise is effective for fitness parameters in older populations [8,9] and adolescent boys [10]. In addition, we showed that low-load push-up exercise training can improve muscle strength and muscle thickness similar to low-load bench-press training using free weights in recreationally active men [11]. However, only one study to date has compared the difference between the effectiveness of online and in-person training interventions using the same bodyweight and resistance-band exercises [12]. “Online training” has been identified as the top and third trending phrase by the American College of Sports Medicine in 2021 and 2022 [13,14]. Considering that online exercise is easily accessible to populations worldwide, our results widely support the validity of online exercise. Although online exercise is undoubtedly useful and important, chronic adaptations to online exercise and the corresponding effect on mental health and well-being remain unclear.

Against this background, this study investigated the effects of online home-based resistance exercise training (through the virtual communication tool Zoom) on fitness, mental health, and well-being. In addition, we replicated the effect of online home-based resistance exercise on fitness, depression, stress, and well-being in comparison to our previous study [12].

## 2. Method

### 2.1. Study Design

This study was conducted from July to September 2020 (Study 1) and July to September 2021 (Study 2) as an intervention-based study. The participants were recruited through social networking services and the Nippon Sport Science University website. In Study 1, sessions were run separately for online and in-person workouts in 2020 and in previous research conducted by the authors [12]. In Study 2, only an online session with the same exercise protocol as Study 1 was conducted. The participants in Studies 1 and 2 were different individuals. We used participant data corresponding to the changes in the Center for Epidemiologic Studies-Depression Scale (CES-D) used in Study 1.

### 2.2. Participants

A total of 67 individuals participated. Of them, 28 participants (13 men and 15 women, average age: 45.1 ± 12.2 years) performed the same exercise training online (*n =* 17), using Zoom (Zoom Video Communications, San Jose, CA, USA) or in person (*n =* 11) in 2020 (Study 1). Additionally, 39 participants (15 men and 24 women; average age: 47.6 ± 10.8 years, Men; height; 169.2 ± 5.1 cm, weight: 66.1 ± 7.9 kg, Women; height: 157.1 ± 4.8 cm, weight: 54.5 ± 6.6 kg) performed eight weeks of online home-based resistance exercise training in 2021 (Study 2). The participants were ≥18 years of age, with or without a chronic illness, and had had no exercise training for at least 6 months prior, and no injuries. In total, 42 individuals participated in Study 2 and three individuals dropped out due to personal reasons. The participants were asked to maintain their normal diet and consistent energy intake throughout the study period; however, as a dietary analysis was not conducted, we could not confirm this aspect. All the participants provided written informed consent. The study was approved by the Ethics Committee of Nippon Sport Science University (020-G04) and was conducted in accordance with the Declaration of Helsinki for Human Research.

### 2.3. Exercise Training

In Studies 1 and 2, the participants were briefed on the purpose of the study, and written informed consent was obtained from them. Before the exercise program, participants attended a training class on the exercises. They performed low-load resistance training (RT) twice a week for eight weeks (16 sessions), with each session lasting 60 minutes and with supervision using Zoom in both Studies 1 [12] and 2. Instructors were graduate students in physical education and the instructor-to-participant ratio was 1:5. Another supervisor responded to any Internet connection issues by telephone. The exercises included nine exercises (leg raise, squat, rear raise, shoulder press, rowing, dips, lunge, single-leg Romanian dead-lift, and push-up). The participants performed body-mass-based RT while standing, sitting on a chair, or using a tube. Each exercise was performed in two or three sets of 10–15 repetitions each session. Rear raise, shoulder press, and rowing were performed using a tube. The sessions were supervised online by students to ensure the correct execution of the exercises. Fitness parameters, isometric and isokinetic strength, cross-sectional area, and augmentation index were measured before and after the eight weeks of training. Post-measurements were performed at intervals between 48–96 h from the final training session.

## 3. Measurements

### 3.1. Mental Health and Well-Being

Psychological well-being was assessed in both pre-and post-exercise training using the following scales:

Well-Being Index (WHO-5). The WHO-5 [15] was administered to measure subjective psychological well-being. The respondents answered five statements about their feelings over the past two weeks. The total score ranged from 0 to 25, with higher scores indicating higher psychological well-being. Scores < 13 indicate poor well-being and suggest depression according to International Statistical Classification criteria [16]. To assess the quality of life, the raw score is multiplied by 4, and the percentage ranging from 0 to 100 (low to a high quality of life) is calculated. A change of ±10 points with 95% confidence intervals has been considered clinically meaningful. The reliability and validity of the WHO-5 are well established [10].

Kessler Psychological Distress Scale (K6). The K6 [17] was administered to measure nonspecific psychological distress. This scale comprises six psychological distress items, such as the tension and restlessness that occurred in the previous month. Response options range from 0 to 4 (none of the time or always), and total scores range from 0 to 24. The scale is highly accurate for diagnosing mood and anxiety disorders (AUC = 0.94) [18]. The optimal cutoff for severe mental disorder was 13+ in the previous study [18].

Center for Epidemiologic Studies-Depression Scale (CES-D). The CES-D [19] uses 20 items to measure the risk for depression and depressive symptoms. It is a 20-item self-report measure that assesses the frequency of current depressive symptoms over the past week. Respondents answer the following frequencies to the 20 questions: most of the time (5 or more days), sometimes (3–4 days), occasionally (1–2 days), and not often (less than 1 day). For each item, a score of 0–3 is applied to each, with higher scores indicating higher frequency. Total scores range from 0 to 60, with clinical-level depressive symptoms associated with scores of 16 or higher. The validity and reliability of the CES-D on the clinical situation have been accepted.

### 3.2. Measurements for a Battery of Tests

To evaluate upper-extremity muscle strength, grip strength was assessed using a Takei hand grip dynamometer (Takei Scientific Instruments, Tokyo, Japan). The participants were allowed two attempts to attain the highest possible rating. A push-up test was performed to evaluate upper-extremity muscle function. Participants were required to touch a 10 cm foam pad placed on the floor with their knees on top of it. Participants were then asked to alternately do push-ups repeatedly for 30 s, and the number of times participants could come to a start position was counted and recorded. The chair stand test was performed to evaluate lower-extremity muscle function. For this test, participants were asked to sit on half of the chair surface with a straightened back and wrists crossed in front of the chest. The participants were then asked to alternately stand and sit repeatedly for 30 s, and the number of times participants could come to a full standing position was counted and recorded. The maximum countermovement vertical jump was assessed using a Yardstick (Swift Performance Equipment Lismore, NSW, Australia). Participants stood side-on to the Yardstick, and keeping their heels on the floor, reached upward as high as possible to displace the zero reference vane. An arm swing and counter-movement jump were used to jump as high as possible with the extended arm displacing the vane at the height of the jump. The sit-and-reach flexibility test is now widely used as a general test of trunk flexibility and was measured with a digital measuring device (Takei Scientific Instruments). Participants were instructed to sit on the floor and stretch their torso and arms out in front of them with their knees straight. Participants were allowed two to three attempts to attain the highest possible rating.

### 3.3. Isometric and Isokinetic Strength

A Biodex System 3 dynamometer (Biodex Medical Systems, Shirley, NY, USA) was used to measure knee extensor strength. Measurements were performed at angular velocities of 60°/s. For knee extensor measurements, the hips were flexed to 90°, with the trunk, waist, and thighs secured with a strap. The center of the knee was aligned with the axis of the dynamometer, with the distal leg secured on a pad. With the knee in an extensor position at 0°, extension from 110° to 0° was performed three times consecutively with maximum contraction.

### 3.4. Analysis of Thigh Cross-Sectional Area (CSA)

The participants were assessed using a 1.5-T whole-body magnetic resonance imaging (MRI) scanner (ECHELON OVAL, Hitachi, Tokyo, Japan). They were instructed to remain as still as possible throughout the process.

The participants were placed in a supine position, and images of the thigh were acquired using a workflow integrated technology (WIT) spine coil 8 and a WIT torso coil. We captured consecutive images of the entire thigh using two-point Dixon imaging. We defined the mid-thigh according to markers attached at the midpoint between the greater trochanter and the lateral condyle of the femur. The image acquisition duration was two minutes. The images of the right thigh were acquired with the following sequences: three-dimensional, repetition time = 13.3 ms; echo time = 6.7 ms and 9.0 ms; flip angle = 60°; optimized field of view = 256 mm × 256 mm; slice thickness = 7 mm; and interslice gap = 0 mm. In-phase and out-of-phase water and fat transaxial images were obtained to create water and fat images for analysis. Dixon images were analyzed using ImageJ software (version 1.44; National Institutes of Health, Bethesda, MD, USA) by a researcher. We measured the CSA of the skeletal muscle of the quadriceps femoris (i.e., the sum of the rectus femoris, vastus lateralis, vastus intermedius, and vastus medialis) at the mid-thigh. Serial axial images were used to identify the muscle boundaries.

### 3.5. Statistical Analyses

SPSS software (version 22; IBM, Armonk, NY, USA) was used for all statistical analyses. We performed pairwise comparisons to determine the groups which differed (paired *t*-test on delta changes). We correlated changes in each parameter of mental health after eight weeks of training sessions and the baseline using Pearson’s correlation. The effect size (ES) was calculated as the absolute pre-test to post-test changes in measurements, such as strength, muscle CSA, fitness, mental health, and well-being divided by the pooled pretest standard deviation (SD) in each group. Statistical significance was set at *p* < 0.05. Data are presented as mean ± SD.

## 4. Results

### 4.1. Study 1

Changes in CES-D after eight weeks of resistance training online and in-person are shown in Table 1. Eight weeks of online RT improved CES-D (ES = 0.41, *p* = 0.003), and a similar tendency was observed in RT with in-person participants (ES = 0.70, *p* = 0.06).

### 4.2. Study 2

The changes in muscle strength, thigh muscle CSA, and fitness parameters, after eight weeks of the RT program are shown in Table 2. The RT program effectively increased vertical jump, grip strength, chair stand test, and push-up test, but there were no differences in the maximal voluntary contraction (MVC), CSA, and sit and reach (Table 2).

There was a significant improvement in CES-D symptoms after the online home-based resistance training in Study 2 (*p* = 0.009). However, there were no significant changes in the WHO-5 and K6. The relationship between the change in the CES-D and the CES-D at baseline (before training) was investigated in this study (Figure 1, r = −0.650, *p* < 0.001). In addition, a similar relationship tendency was shown in WHO-5 (r = 0.290, *p* = 0.069) and K6 (r = −0.200, *p* = 0.231).

## 5. Discussion

This study investigated the effect of online home-based resistance exercise training at home (through the virtual communication tool—Zoom) on fitness, depression, stress, and well-being. The primary findings of our study are as follows: (1) eight weeks of online home-based resistance exercise improved the mental health of those who scored higher on CES-D; and (2) physical fitness improved during the online home-based resistance exercise intervention. These findings suggested that the effect of online home-based resistance exercise may play a role in improving fitness and is one of the non-pharmacological therapies for mental health well-being. In addition, the findings replicated those of a previous study conducted by the authors [12].

Additionally, an extensive study of the literature indicates that this study makes a novel contribution by clarifying the effect of online supervised resistance training on fitness, mental health, and well-being. Our study suggested that online supervised resistance training improved CES-D in healthy individuals but did not significantly impact the WHO-5 and K6. Furthermore, the results can be replicated in the improvement of CES-D using the same exercise protocol of supervised online training used in Study 1 and Study 2. Interestingly, a higher effect size was observed when training in person (ES = 0.70) compared to online training (ES = 0.41) in Study 1. However, the sample size was small (*n* = 11; in-person training). A future study using a randomized controlled trial in a large sample size is necessary to reveal the effect of online or in-person resistance training on mental health responses.

The results of our previous study indicated a similar training response to resistance training in both online and in-person group settings [12]. Similarly, Carlson et al. [20] employed a cross-over design to study the effects of three weeks of supervised one-to-one traditional resistance exercise sessions (in person) and a virtual personal training protocol performed using body-mass-based training in 20 trained participants (13 men and seven women). The results suggested that short-term supervised virtual resistance training is as efficacious as traditional supervised studio-based resistance training.

The results of this study indicated that low-load resistance training improved depressive symptom status even in the case of fewer changes in fitness parameters and muscle size after exercise training compared with a previous study (Anonymous, 2021). In addition, there was a negative correlation between baseline and changes in CES-D. It means that online home-based resistance training is more valid for participants with high depressive symptoms status.

A study by Penninx et al. [21] investigated the effect of resistance and aerobic training on changes in CES-D and demonstrated a declining trend in depressive symptoms in both exercises. This study was in line with our observational findings that about half of the older persons with high CES-D scores significantly improved their depressive symptomatology over a short follow-up period. Dieli-Conwright et al. [22] reported that 16 weeks of moderate to high intensity aerobic and resistance exercise showed greater improvement of CES-D (from 15.1 ± 3.3 to 9.7 ± 2.5) than controls in breast cancer patients. High-intensity resistance training showed trends for reduced depressive symptoms (from 15.43 ± 7.49 to 13.78 ± 8.02) at a six-month follow-up [23].

However, our results showed that there was no significant difference in the WHO-5 and K6 after eight weeks of online training in Study 2. One possible reason for this result could be that the participants were relatively healthy. In this study, no participants exceeded the criterion in the K6 at baseline and only three participants, out of 39, had poor well-being (<13) using the WHO-5. Ejiri et al. [24] suggested that exercise as a health-promoting coping behavior during the stay-at-home order was associated with psychological well-being (WHO-5) in Japan’s older population. Especially, walking for maintaining physiological health was reported as effective for mental health. This was in line with the findings of previous studies on COVID-19 [25]. Thus, it is possible that there was not enough in terms of characteristics in participants and exercise modalities (resistance or aerobic exercise) to change the K6 and WHO-5 measurements.

Resistance exercise training improves depressive symptoms due to probable associations with metabolites after resistance training. The plausible mechanism is the secretion of biomarkers due to muscle contraction, such as insulin-like growth factor 1 (IGF-1) and brain-derived neurotrophic factor (BDNF). Previous studies showed a strong association between depressive symptoms and cognition among older adults with lower levels of circulating IGF-1 [26]. The level of IGF-1 was simultaneously increased in both blood and the hippocampus in animals trained with resistance exercise [27]. These results were at least partially consistent with human studies that show an increase in serum IGF-1 after resistance exercise [28,29]. On other hand, low serum BDNF levels were observed in patients with major depression [30,31]. Additionally, future research needs to examine the mechanism of improvement of depressive symptoms in online supervised resistance training.

While this study has novel implications for the literature and practice, it presents the following limitations that can be addressed in future research. First, even as online exercise and training are popular, this form of exercise needs to be standardized; future studies should include wider age groups and statuses (for example: untrained and trained participants) to extend the study scope. Second, the participants were asked to maintain their normal diet and consistent energy intake throughout the study period; however, as a dietary analysis was not conducted, we could not confirm the effect of a change in diet in this study. We recommend that future studies consider dietary analyses. Third, we could not correct the biomarker during the examination. To clarify the mechanisms between resistance exercise and the improvement of mental health, one may need to assess the biomarkers and signaling molecules produced by skeletal muscle contraction. Fourth, selection bias may have remained because of the study design, which was a non-randomized intervention study. To rigorously evaluate the effects of online home-based resistance exercise on mental health, it may be necessary to conduct a randomized controlled trial.

## 6. Conclusions

In conclusion, our results suggested that online low-load resistance training improved fitness parameters and depressive symptoms (CES-D) in healthy middle-aged populations. Therefore, our findings indicated that online home exercise training using video communication may be effective as one of the recommended exercise regimes for homebound individuals and special populations, including higher-risk individuals who need to stay at home during the COVID-19 pandemic.

## Figures and Tables

**Figure 1 ijerph-20-01769-f001:**
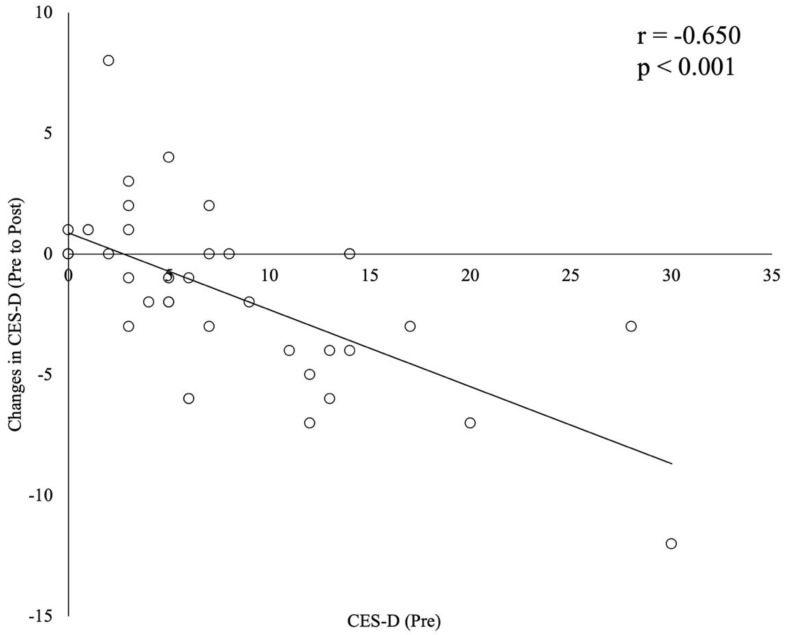
A correlation between baseline and changes in CES-D after eight weeks of online low load resistance training in Study 2.

**Table 1 ijerph-20-01769-t001:** Changes in the CES-D, K6, and WHO-5 after 8 weeks of online or in-person resistance training in Studies 1 and 2.

		Pre		Post		Mean Difference	Effect Size (95%CI)	*p*-Value
		Value	Reaching the Criteria (*n*)	Value	Reaching the Criteria (*n*)			
Study 1								
CES-D	Online (*n* = 17)	9.6 ± 7.2	3	6.9 ± 6.1	1	−2.7 ± 3.1	−0.41 (−1.08, 0.28)	0.003
	In person (*n* = 11)	8.2 ± 4.4	0	5.5 ± 3.1	0	−2.6 ± 4.1	−0.70 (−1.53, 0.19)	0.060
Study 2								
CES-D (*n* = 39)		7.7 ± 7.2	4	6.1 ± 5.6	2	−1.6 ± 3.6	−0.24 (−0.69, −0.21)	0.009
WHO-5 (*n* = 39)		74.3 ± 16.3	5	76.2 ± 17.0	4	2.0 ± 11.6	0.12 (−0.33, 0.56)	0.302
K6 (*n* = 39)		1.6 ± 2.0	0	1.4 ± 2.1	0	−0.2 ± 1.2	−0.09 (−0.53, 0.36)	0.343

CES-D; Center for Epidemiologic Studies-Depression Scale, WHO-5; The World Health Organization-Five Well-Being Index, K6; Kessler Psychological Distress Scale, 95%CI; 95% confidence interval. Reaching the criteria: SEC-D > 16, WHO-5 < 13, K6 > 11.

**Table 2 ijerph-20-01769-t002:** Changes in fitness parameters and muscle cross-sectional area in Study 2 (*n =* 39).

	Pre	Post	Mean Difference	Effect Size (95%CI)	*p*-Value
MVC, N·m (Right)	167.4 ± 50.5	170.9 ± 50.1	3.4 ± 26.0	0.07 (−0.39, 0.52)	0.426
MVC, N·m (Left)	156.6 ± 52.5	159.6 ± 46.3	3.0 ± 23.8	0.06 (−0.38, 0.50)	0.431
MVC, N·m (Average)	160.6 ± 50.6	164.0 ± 46.3	3.4 ± 21.9	0.07 (−0.37, 0.51)	0.34
CSA, mm^2^	5098.3 ± 1118.3	5147.3 ± 1140.2	49.0 ± 227.5	0.04 (−0.43, 0.51)	0.212
Vertical jump, cm	32.8 ± 8.1	33.9 ± 7.9	1.1 ± 3.1	0.14 (−0.32, 0.59)	0.038
Grip strength, kg (Right)	30.6 ± 7.8	31.7 ± 7.8	1.2 ± 2.2	0.15 (−0.30, 0.59)	0.003
Grip strength, kg (Left)	30.6 ± 7.2	30.4 ± 7.7	−0.2 ± 3.0	−0.03 (−0.48, 0.41)	0.611
Grip strength, kg (Average)	30.6 ± 7.4	31.1 ± 7.6	0.5 ± 2.0	0.06 (−0.38, 0.50)	0.154
Chair stand test, times	27.7 ± 5.0	28.9 ± 4.5	1.2 ± 3.8	0.26 (−0.19, 0.70)	0.048
Push up test, times	16.0 ± 9.2	20.2 ± 9.5	4.1 ± 4.5	0.44 (−0.03, 0.90)	<0.001
Sit and reach, cm	42.8 ± 8.8	43.3 ± 9.4	0.6 ± 4.3	−0.07 (−0.51, 0.38)	0.391

Mean ± S.D., MVC; maximal voluntary contraction, CSA; cross-sectional area, 95%CI; 95% confidence interval.

## Data Availability

The datasets used and/or analyzed during the current study are available from the corresponding author upon reasonable request.

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
