# Peer review of "Effect of Online Home-Based Resistance Exercise Training on Physical Fitness, Depression, Stress, and Well-Being in Middle-Aged Persons: A Pilot Study"

_ijerph, 2023, doi:10.3390/ijerph20031769_

Round 1

Reviewer 1 Report

The paper investigated the effect of online home-based resistance exercise training on physical fitness, mental health, and well-being in middle-aged persons.

 1.      the article does not highlight the selection criteria of the study group(s) - Sample Inclusion/Exclusion Criteria - and this creates confusion

2.      paper does not explain adequately if the study group(s) applied the same set of exercises and set-up as a previous successful (supervised) training program.

3.      it is not very clear if, after these online exercises, an interview was conducted with the people in the group regarding the evaluated/analysed aspects - how many accepted and how many refused

4.      the people in the study (11) who did onsite exercises, how do they influence the results and the interpretation of the results in the context in which the online "approach" is evaluated

5.      How do you deal with the time delay issue of the on-line system? – please address this

6.      What was the reaction of the target group in situations where Zoom did not work, connection errors occurred, etc. and how these situations influenced the “patient's” well-being – please address this

7.      Please briefly detail in the conclusions what does it add to the subject area compared with other published material?

Author Response

We are grateful to Reviewer for the detailed review and useful comments. As indicated in the responses that follow, we have taken care to address all these comments in the revised version of our paper. We hope that the revisions in our paper and our responses to the comments are satisfactory.

he paper investigated the effect of online home-based resistance exercise training on physical fitness, mental health, and well-being in middle-aged persons.

1.the article does not highlight the selection criteria of the study group(s) - Sample Inclusion/Exclusion Criteria - and this creates confusion

=====

we added information of inclusion: Adults ≥18 years of age with or without a chronic illness, no exercise training at least 6 month, no injury

  1. paper does not explain adequately if the study group(s) applied the same set of exercises and set-up as a previous successful (supervised) training program.

======

We revised the sentence and add sentence “Instructors were university educated physical education teachers and the instructor to participant ratio was 1:5.”

  1. it is not very clear if, after these online exercises, an interview was conducted with the people in the group regarding the evaluated/analysed aspects - how many accepted and how many refused

====

We have added information.

  1. the people in the study (11) who did onsite exercises, how do they influence the results and the interpretation of the results in the context in which the online "approach" is evaluated

====

Subjects do not know the purpose of the study. I don't think there will be any particular impact.

  1. How do you deal with the time delay issue of the on-line system? – please address this 

====

Another supervisor responded to the connection trouble by telephone.

  1. What was the reaction of the target group in situations where Zoom did not work, connection errors occurred, etc. and how these situations influenced the “patient's” well-being – please address this

====

We did a connection test before intervention, so there were few connection problems.

  1. Please briefly detail in the conclusions what does it add to the subject area compared with other published material?

===

We have added the information in conclusion

Reviewer 2 Report

Review of the manuscript entitled: Effect of Online Home-Based Resistance Exercise Training on Physical Fitness, Mental Health, and Well-Being in Middle-Aged Persons. The manuscript submitted is appropriate to the subject matter and scientific rigor. Some remarks improving the quality of future research. and suggested changes and comments to the submitted manuscript in order to improve the quality of the planned research and future publications below:

1. Introduction, line 15 : Please enter the number of people who took part in the experiment counts do not match: 16 men and 18 women = 34 not 28 would you please check it. 

2. The material of the work was poorly described - no quantitative differentiation in terms of body weight (men and women  separately), BMI, height etc. Lack of exclusion criteria and qualification for the experiment.

3. In the article cited by the authors [1] of the reviewed article, we read: "The results show that the lockdown restrictions did not lead to decrease in exercise levels. Also, those who exercised more frequently during the pandemic" in the introduction to the peer-reviewed article we can read “which lead to decreased physical activity.” Please change it or provide other references..

4. Line: 47 i 48. The authors refer to references [1] and give others [2]

5. Line 53: Please list and describe these views, not just give references [3.4]

6. Line 66: (Anonymous, 2021) This is reference I can not find it in bibliography

7. In the description of the methods, it is necessary to specify how the load was selected during the training and how it was controlled, and whether the intensity of the classes changed during the year, and not only "The participants performed low-load RT"

8. References should be provided when describing tests (line 137-149)

9. The description of the results describes the Verdical jump and Push up test but this is not described and methodology by authors "Study 2 The changes in muscle strength, thigh muscle CSA, fitness parameters, after 8 weeks 3 of RT program are shown in Table 2. The RT program effectively increased vertical jump".

10. In discussion  authors write „Resistance exercise training improves depressive symptoms due to probable associ- 64 ation with metabolites after middle or high intensity resistance training”. But they describing  measurements for battery of tests   they write only about „The participants performed low-load RT”. 

11. In Reference  [9]   „Thompson WR. Worldwide Survey of Fitness Trends for 2022. ACSM's Health & Fitness Journal ...... "would you please correct the name author " Walter R. Thompson" 

12. The discussion 'results' chapter and has a weak correlation with work results. For example, there is no description of the test results, there is no information about the results of the Verdical jump and Push up test.

Author Response

We are grateful to Reviewer  for the detailed review and useful comments. As indicated in the responses that follow, we have taken care to address all these comments in the revised version of our paper. We hope that the revisions in our paper and our responses to the comments are satisfactory.

Review of the manuscript entitled: Effect of Online Home-Based Resistance Exercise Training on  Physical Fitness, Mental Health, and Well-Being in Middle-Aged Persons. The manuscript submitted is appropriate to the subject matter and scientific rigor. Some remarks improving the quality of future research. and suggested changes and comments to the submitted manuscript in order to improve the quality of the planned research and future publications below:

1. Introduction, line 15 : Please enter the number of people who took part in the experiment counts do not match: 16 men and 18 women = 34 not 28 would you please check it. 

=====

Thank you for your comment. It was our mistake. We collected them

2. The material of the work was poorly described - no quantitative differentiation in terms of body weight (men and women  separately), BMI, height etc. Lack of exclusion criteria and qualification for the experiment.

======

We add the exclusion criteria and information of weight and height in both men and women

3. In the article cited by the authors [1] of the reviewed article, we read: "The results show that the lockdown restrictions did not lead to decrease in exercise levels. Also, those who exercised more frequently during the pandemic" in the introduction to the peer-reviewed article we can read “which lead to decreased physical activity.” Please change it or provide other references..

=====

We changed sentence and reference.

4. Line: 47 i 48. The authors refer to references [1] and give others [2]

=====

We have revised this paragraph.

5. Line 53: Please list and describe these views, not just give references [3.4]

======

We have revised this paragraph.

6. Line 66: (Anonymous, 2021) This is reference I can not find it in bibliography

====

We added the reference in text.

7. In the description of the methods, it is necessary to specify how the load was selected during the training and how it was controlled, and whether the intensity of the classes changed during the year, and not only "The participants performed low-load RT"

====

The participants performed body weight-based RT while standing, sitting on a chair, or using a tube. Each exercise was performed 2 or 3 sets of 10-15 repetitions in session.

8. References should be provided when describing tests (line 137-149)

====

In present study, we don’t have specific or original test.

9. The description of the results describes the Verdical jump and Push up test but this is not described and methodology by authors "Study 2 The changes in muscle strength, thigh muscle CSA, fitness parameters, after 8 weeks 3 of RT program are shown in Table 2. The RT program effectively increased vertical jump".

====

We added the information of the Vertical jump and Push up test

10. In discussion  authors write „Resistance exercise training improves depressive symptoms due to probable association with metabolites after middle or high intensity resistance training”. But they describing  measurements for battery of tests   they write only about „The participants performed low-load RT”. 

====

We revised the session.

11. In Reference  [9]   „Thompson WR. Worldwide Survey of Fitness Trends for 2022. ACSM's Health & Fitness Journal ...... "would you please correct the name author " Walter R. Thompson" 

====

We have corrected.

12. The discussion 'results' chapter and has a weak correlation with work results. For example, there is no description of the test results, there is no information about the results of the Verdical jump and Push up test.

======

We have added the information of Vertical jump and Push up test

Reviewer 3 Report

It is a valuable attempt to examine the effect of home-based resistance exercise training to reduce stress responses and improve mental health and well-being. But, I think it will be necessary to supplement much it to be published in the Q1 or Q2 journal. Those things needed to be revised are as follows.

1. In the introduction, a rationale should be provided as to which elements of or how applied exercise training can reduce stress and depression and promote well-being. And, it is necessary to state more convincingly that such a study is necessary.

2. Research procedures or experimental procedures need to be described in detail independently from the description of participants.

3. Stress and depression cannot represent mental health, so it is necessary to specify stress and depression in the title of the article.

4. Results of the study do not appear to be clinically significantl. What do you think?

5. What was the purpose of analyzing the correlations between pre- and post-depression scores and between well-being (WHO-5) and stress (K6)? I think these analyses is meaningless.

6. Are the variables in normal distribution enough to perform parametric analysis?

7. The biggest weakness of this study is that there is no control or comparison group, so it is difficult to conclude that the treatment is effective. It can be described as a limitation of the study, but since it is not a difficult sample to obtain, a control group should have been established to ensure internal validity.

8. Please, pay more attention to to structuring and editing the manuscript. For example, a list of abbreviations should go to the back of the manuscript.

Author Response

We are grateful to Reviewer  for the detailed review and useful comments. As indicated in the responses that follow, we have taken care to address all these comments in the revised version of our paper. We hope that the revisions in our paper and our responses to the comments are satisfactory.

It is a valuable attempt to examine the effect of home-based resistance exercise training to reduce stress responses and improve mental health and well-being. But, I think it will be necessary to supplement much it to be published in the Q1 or Q2 journal. Those things needed to be revised are as follows.

  1. In the introduction, a rationale should be provided as to which elements of or how applied exercise training can reduce stress and depression and promote well-being. And, it is necessary to state more convincingly that such a study is necessary.

=====

Thank you for your comment. We revised the introduction.

  1. Research procedures or experimental procedures need to be described in detail independently from the description of participants.

=====

We have added the study design section.

  1. Stress and depression cannot represent mental health, so it is necessary to specify stress and depression in the title of the article.

  ====

We changed our title of the article.

  1. Results of the study do not appear to be clinically significantl. What do you think?

====

Our findings indicate that online home exercise training using video communication might be effective as one of the recommended exercise regimes for homebound individuals and special populations who need to stay at home, higher risk individuals, or low socioeconomic status individuals during the COVID-19 pandemic.

  1. What was the purpose of analyzing the correlations between pre- and post-depression scores and between well-being (WHO-5) and stress (K6)? I think these analyses is meaningless.

====

Thank you for your suggestion. We show the association between pre data and changes (pre-post) in CESD, WHO-5 and K6. These results indicated the effect of resistance exercise training on each score is higher change in individuals with poor scores than individuals with better scores.

  1. Are the variables in normal distribution enough to perform parametric analysis?

====

This data is normally distributed.

  1. The biggest weakness of this study is that there is no control or comparison group, so it is difficult to conclude that the treatment is effective. It can be described as a limitation of the study, but since it is not a difficult sample to obtain, a control group should have been established to ensure internal validity.

=====

Thank you for your comment. We agree with reviewer. We added limitation in this study.

  1. Please, pay more attention to to structuring and editing the manuscript. For example, a list of abbreviations should go to the back of the manuscript.

=====

Thank you. We revised our manuscript

Reviewer 4 Report

Dear Authors,

Thank you for the opportunity to review this interesting paper. In my opinion, the publication considers an interesting and extremely topical research problem and the conclusions are of significant application and public health importance. The conclusions of the publication indicate that online low-load resistance training improves fitness parameters and depressive symptoms. Additionally, home exercise training using video communication might be effective as one of the recommended exercise regimes for homebound individuals and special populations, including higher-risk individuals who need to stay at home during the COVID-19 pandemic. From this perspective, the presented results are highly useful. However, the method of presenting the methods, selection of research material and results is very unsatisfying. In addition, the small size of the group and the descriptions of the limitations of the research suggest that there are pilot studies.

Comments and Suggestions for Authors:

1. The research material is small and requires detailed elaboration. Please provide the criteria for inclusion and exclusion of the subjects in the study, the minimum and maximum age ranges of the subjects, basic somatic parameters (height, weight, BMI), and any other data characterizing the study group.

2. In the section on the statistical methods used, it is necessary to describe in more detail what data has been provided. Please prove that parametric tests (Pearson correlations) can be used.

3. The presentation of the results needs to be clarified, e.g. 

-in the tables (in the upper line, i.e. in the table header) it is necessary to specify precisely what results are presented in the subsequent columns; 

-please give the results in the columns with the same accuracy (refers to the number of places after the full stop); 

-in the effect size column, enter commas in brackets; 

-p=0.06 means that the result is not significant.

4. Please check and edit the text of the manuscript carefully: 

-the method of notation of results should be unified - using or not spaces before and after the "=" sign; 

-the record of literature and the method of citation should be standardized, for example (Anonymous, 2021).

5. ICD-11 is currently in force, not ICD-10 - please verify.

6. Due to the small size of the material and the relevant provisions in the limitation study section, please consider introducing the term - pilot study - into the title of the manuscript, if indeed such is the case.

7. In my opinion the discussion section is well written. 

8. Thank you for the important information about the aesthetic consent for the research.

Thank You.

Author Response

We are grateful to Reviewer  for the detailed review and useful comments. As indicated in the responses that follow, we have taken care to address all these comments in the revised version of our paper. We hope that the revisions in our paper and our responses to the comments are satisfactory.

Thank you for the opportunity to review this interesting paper. In my opinion, the publication considers an interesting and extremely topical research problem and the conclusions are of significant application and public health importance. The conclusions of the publication indicate that online low-load resistance training improves fitness parameters and depressive symptoms. Additionally, home exercise training using video communication might be effective as one of the recommended exercise regimes for homebound individuals and special populations, including higher-risk individuals who need to stay at home during the COVID-19 pandemic. From this perspective, the presented results are highly useful. However, the method of presenting the methods, selection of research material and results is very unsatisfying. In addition, the small size of the group and the descriptions of the limitations of the research suggest that there are pilot studies.

Comments and Suggestions for Authors:

  1. The research material is small and requires detailed elaboration. Please provide the criteria for inclusion and exclusion of the subjects in the study, the minimum and maximum age ranges of the subjects, basic somatic parameters (height, weight, BMI), and any other data characterizing the study group.

======

We added the criteria and basic data in subjects.

  1. In the section on the statistical methods used, it is necessary to describe in more detail what data has been provided. Please prove that parametric tests (Pearson correlations) can be used.

=====

This data is normally distributed. We have added the sentence in statistical analysis section.

  1. The presentation of the results needs to be clarified, e.g. 

-in the tables (in the upper line, i.e. in the table header) it is necessary to specify precisely what results are presented in the subsequent columns; 

-please give the results in the columns with the same accuracy (refers to the number of places after the full stop); 

-in the effect size column, enter commas in brackets; 

-p=0.06 means that the result is not significant.

=====

We corrected the results (including table 1, 2)

  1. Please check and edit the text of the manuscript carefully: 

-the method of notation of results should be unified - using or not spaces before and after the "=" sign; 

-the record of literature and the method of citation should be standardized, for example (Anonymous, 2021).

=====

Thank you we revised our manuscript according to your comment.

  1. ICD-11 is currently in force, not ICD-10 - please verify.

====

Thank you.

  1. Due to the small size of the material and the relevant provisions in the limitation study section, please consider introducing the term - pilot study - into the title of the manuscript, if indeed such is the case.

====

We added “a pilot study” in the title

  1. In my opinion the discussion section is well written. 
  2. Thank you for the important information about the aesthetic consent for the research.

====

Thank You for your comment.

Round 2

Reviewer 3 Report

I think the manuscript has been supplemented based on the comments in the first review. So it's worth it as a pilot study.  Even so, it would be better if you try to improve the quality of your paper overall.  Thank you for your hard work.

Author Response

Thank you for reviewing our manuscript.
We are grateful that your comments have improved the manuscript.

Reviewer 4 Report

Dear Authors,

Thank you for the opportunity to review this paper again.

The text of the publication has been improved. 

Thank you very much for this additions. This is exactly what should have been added to the manuscript.

I am satisfied with the revised version of the manuscript.

Thank you for the opportunity to review this article.

Author Response

(The authors gave the same response as above.)
